# miRNA Library Preparation Optimisation for Low-Concentration and Low-Volume Paediatric Plasma Samples

**DOI:** 10.3390/ncrna11010011

**Published:** 2025-02-05

**Authors:** Oenone Rodgers, Chris Watson, Thomas Waterfield

**Affiliations:** Wellcome-Wolfson Institute for Experimental Medicine, Queen’s University Belfast, Belfast BT9 7BL, UK

**Keywords:** miRNA, paediatric, biofluids, nucleic acid diagnostics, infection, biomarker, miRNA library preparation, miRNA sequencing, optimisation, protocol

## Abstract

**Background:** Analysing circulating miRNAs in paediatric plasma is challenging due to typically low sample volumes. The QIAseq miRNA UDI Library Kit (Qiagen, Hilden, Germany) was selected as it has a proven track record with a specific protocol for plasma and serum. The protocol, however, required optimisation for use with low-volume paediatric plasma samples before generating acceptable yields in our cohort. **Methods:** The miRNeasy Serum/Plasma kit (Qiagen) and the MagMAX miRVana Total Isolation kit (ThermoFisher Scientific, Waltham, MA, USA) were assessed following the manufacturer’s instructions with 100 µL and 200 µL of paediatric plasma. Libraries were prepared using the QIAseq miRNA UDI Library Kit (Qiagen). Optimisations were made for the QIAseq miRNA UDI Library Kit (Qiagen) using total RNA extracted with the miRNeasy Serum/Plasma kit (Qiagen) from 100 µL of plasma. **Results:** Prior to optimisation, both RNA extraction kits underperformed with the QIAseq miRNA UDI Library kit, producing low miRNA library yields ranging between 0 and 1.42 ng/µL. Plasma input volumes of 100 µL and 200 µL demonstrated no significant differences. Adjusting the QIAseq protocol for low RNA concentrations improved miRNA library yields, an average of 5.6 ng/µL and a maximum of 24.3 ng/µL across 92 samples. The optimised protocol showed no age or gender biases with the QIAseq kit. **Conclusions:** Failure rates in miRNA library preparations are rarely reported, making it hard to gauge whether the 8.7% failure rate observed here is typical. However, given the challenges of using low-concentration, low-volume paediatric plasma, this represents a significant improvement over previous attempts, supporting further research in the field.

## 1. Introduction

Small RNAs, which include microRNAs (miRNAs), small interfering RNAs (siRNAs), PIWI-interacting RNAs (piRNAs), and transfer RNA-derived small RNAs, are defined as having less than 200 nucleotides [1]. miRNAs, siRNAs, and piRNAs play crucial roles in regulating gene expression at multiple levels, including post-transcriptional and translational processes, and can significantly influence epigenetic modifications [1]. Among all small RNA subtypes, miRNA research is more advanced than other small RNAs [2]. While other small RNAs like siRNAs and piRNAs have their own biological importance, the combination of established research tools, clinical relevance, and practical considerations makes miRNAs the primary focus in small RNA sequencing studies.

Since their discovery in the early 1990s [3], interest in miRNAs has grown steadily; they are involved in a wide range of biological processes, and profile alterations are found in many diseases [4], including malignancies [5] and infectious diseases [6]. In humans, miRNA can be detected in a variety of different sample types, including biofluids such as plasma, serum, saliva, and urine [6]. Circulating miRNAs are viewed as promising biomarker candidates due to their stability in plasma and serum, which is attributed to their association with protein complexes like AGO2 and their encapsulation within exosomes [6,7,8,9,10]. Additionally, miRNAs exhibit high specificity and sensitivity early in disease processes [5,10], which may improve diagnostic abilities across many different types of disease.

Methods for detecting miRNA in plasma and serum include RT-qPCR, microarray, and next-generation sequencing (NGS) [11]. The choice depends on prior knowledge and project goals. RT-qPCR, a “closed hypothesis” method, requires specific probes for miRNAs of interest. Microarray, a “moderately open” method, analyses thousands of miRNAs but still relies on predefined probes. NGS offers the most “open hypothesis”, enabling unbiased, comprehensive analysis and novel miRNA discovery. To date, over 2600 mature miRNAs have been identified in the human genome, with numbers increasing through small RNA sequencing [4]. However, the application of miRNA NGS in paediatric research is challenging due to the low volume and low RNA concentration of plasma and serum samples available from young, often unwell patients. Typically, plasma or serum samples are often favoured in the field of clinical miRNA detection, as it is a relatively non-invasive method to access biomarkers [12]. Whole blood is usually avoided because the cellular material is rich in miRNA, which can interfere with the overall miRNA profiles detected [13]. Even with plasma and serum samples, analysis is complicated by high levels of inhibitors, such as RNAses, and relatively low RNA concentrations [9].

Several other factors can influence the quality and quantity of miRNA from plasma and serum, such as the type of RNA extraction kit and the volume of sample used to extract RNA [13]. Different methods are available to extract and purify miRNA from plasma and serum samples, including Trizol methods and magnetic bead isolation methods. Trizol reagent is used to denature protein primarily bound to nucleic acids and inhibit RNAse activity, preserving the RNA component of the sample [14]. Although the Trizol method is the most widely used method for RNA extraction, it is time-consuming and can introduce contaminants such as DNA [15]. Due to lengthy processing times, there is also an increased risk of RNA degradation, leading to a decreased final yield of pure total RNA [15]. Furthermore, Trizol extraction methods are associated with a selective loss of small RNA molecules with a low GC content [16]; however, this can be mitigated by using biofluid-specific small RNA extraction kits [6]. Alternatively, magnetic particles can be used to isolate total RNA. This technique has several advantages over the Trizol method, such as being able to extract from whole blood, plasma, and serum, and being scaled to any sample volume [15]. The sample processing is also less intensive than with the Trizol method, potentially reducing nucleic acid degradation, which could further reduce elution yield [15].

Conducting miRNA NGS studies with clinical samples is difficult, as a substantial amount of serum or plasma must be extracted to achieve adequate RNA yield for successful sequencing [7]. Quantification bias, which is particularly prominent in biofluids due to low miRNA concentration, can occur due to unequal ligation efficiencies leading to over- and underestimation of the true miRNA signatures [17]. Low input RNA concentrations perform poorly during library preparation, where increased adapter dimers, poor yield, and increased non-miRNA reads can be expected [9]. This is especially relevant in paediatrics, where obtaining large volumes of blood can be difficult due to young age or severe illness. As a result, pooled samples are frequently used, which can reduce the statistical power required for clinical research and increase the risk of false-negative results [7]. However, using pooled samples in biomarker discovery can be beneficial, as only the most robust signals are likely to remain, while weaker signals that are unlikely to withstand further validation are filtered out early in the process.

The QIAseq miRNA UDI Library Kit has been reported to perform well, demonstrating the highest miRNA diversity from a fixed number of reads mapping to miRNAs, and it correlates most closely with RT-qPCR results with human plasma [9,18]. The QIAseq kit is designed to reduce bias, minimise adaptor dimer formation, and decrease contamination from non-miRNAs, thereby enabling effective use of low RNA inputs through optimised reaction chemistry [9]. A systematic assessment of six low-input miRNA library preparation kits found the QIAseq kit to reliably demonstrate top performance among those tested [18]. Furthermore, a separate assessment of seven commercially available small RNA sequencing library preparation kits for biofluids found the QIAseq kit to perform among the best in all metrics tested [17]. The QIAseq handbook offers reagent ratios and PCR amplification cycles specifically for serum and plasma samples, where it assumes a 10 ng input RNA concentration. Other reagent dilutions are offered throughout the protocol for 500 ng, 100 ng, 10 ng, and 1 ng of total RNA input. Notably, the QIAseq kit also incorporates unique molecular indices (UMIs) into each cDNA molecule, allowing for post-library production correction of PCR bias.

The QIAsq kit, however, has not been optimised for use with low-volume paediatric samples. Consequently, the QIAseq kit was selected for optimisation with paediatric samples due to its high likelihood of generating reliable and high-quality results in plasma. The successful application of small RNA sequencing in paediatric studies, however, is frequently impeded by the low RNA yield and small sample volumes typical of paediatric patients. Standard sequencing protocols often fail to generate accurate and reproducible data under these conditions, limiting the utility of this approach in paediatric research. This project aims to resolve some of these issues in generating viable libraries with low-volume and low-concentration samples. By improving sensitivity and reproducibility under low-input conditions, this research aims to facilitate the discovery of novel miRNA biomarkers and enable more robust biomarker discovery and insights into paediatric health and disease.

## 2. Results

### 2.1. Testing RNA Extraction Kit and Starting Volume of Plasma in Optimisation 1

The libraries underwent quality control via the Fragment Analyzer, which assesses the concentration of the miRNA library itself, found with a peak of ~200 bp, and the purity of the sample produced. Varying library concentrations were detected in the samples using Optimisation 1, with all the miRNA libraries being very low in concentration (Figure 1a). The MagMAX RNA extraction kit produced a range in miRNA library concentration of 0–0.11 ng/µL and an average of 0.027 ng/µL. The miRNeasy RNA extraction kit produced a higher range in miRNA library concentration of 0–1.42 ng/µL and an average of 0.301 ng/µL. The highest yield was in the inflammatory pool at a 100 µL starting volume of plasma at 1.415 ng/µL from the miRNeasy kit. An example of the graph produced via the Fragment Analyzer (Figure 1b) highlights the undistinguished peak at approximately 200 bp but a prominent peak at approximately 50–60 bp. Representing unbound adaptors, a peak at 50–60 bp indicates incomplete reactions, and coupled with the extremely low miRNA library yield, directed the investigation towards optimisation in these samples.

Although many of the miRNA library yields were poor for these samples, it was noted that all the samples from the <90 days pooled group failed to generate any library at all. It was unclear whether this was due to extremely low RNA concentration in the samples, which may prove to be an inherent issue with using samples from young infants, or if this was due to inhibitor interference. All the pooled samples used in the library preparation and additional individual samples were assessed via qPCR with a known quantity of the spike-in miRNA UniSp6 (Figure 2). The high Ct value and wide variation for the <90 days (Seq) samples, however, suggest inhibitor interference, hindering library formation. These high levels of inhibitors interfere with both endogenous and exogenous miRNAs alike, therefore also affecting UniSp6.

There was no significant difference in the total reads produced during sequencing between the two RNA extraction kits tested (Figure 3a). However, one sample extracted via the MagMAX kit failed to produce a library. The mean total reads were higher in the samples extracted with the miRNeasy kit (1.89 × 10^7^ total reads) compared with the MagMAX kit (5.76 × 10^6^ total reads). The percentage of unique reads (%UMI) also had a greater mean in the miRNeasy kit-extracted samples (6.7%) compared with the MagMAX (2.7%), although the overall differences between the kits were non-significant (Figure 3b).

Both the miRNeasy and MagMAX miRVana RNA extraction kits have been optimised by the manufacturer for use with a 200 µL starting volume of plasma. To assess the impact on sequencing and if it was even viable to run successful experiments with 100 µL volume, both 100 µL and 200 µL were compared via small RNA sequencing. No significant difference was found between samples extracted using 100 µL or 200 µL of plasma (Figure 4a). There was a higher mean total number of reads for samples extracted from 100 µL of plasma (1.67 × 10^7^ total reads) compared to 200 µL (7.95 × 10^6^ total reads). Additionally, the percentage of unique reads of total reads was non-significant between the two volumes (Figure 4b). The mean values of %UMI reads were uniform between 100 µL (4.7%) and 200 µL (4.6%). However, there was an outlier in 200 µL, showing a much greater percentage of unique reads in one sample. Nevertheless, the 100 µL extracted samples appear to have a more consistent and precise %UMI compared to 200 µL, indicating expected results may be more consistent.

### 2.2. Assessing Optimisations

To improve the miRNA library yield from these low-concentration and low-volume samples, three sets of alterations to the protocol were assessed to see if they were beneficial for the samples. All three optimisations were beneficial to the formation of a miRNA library (Figure 5). Optimisation 2 provided the highest yield for individual samples extracted with 100 µL of plasma, with an average miRNA library of 1.6 ng/µL. This result was achieved by condensing the total RNA elution from 10 µL to 5 µL. Therefore, in a 10 µL uncondensed sample, there would be approximately 1 ng of total RNA. Consequently, in a 5 µL uncondensed standard elution, approximately 0.5 ng of total RNA could be expected when extracted from 100 µL of paediatric plasma. An RNA input of 0.5 ng is too low a concentration for the kit handbook-recommended reagent dilutions and PCR amplification cycles, as the lowest it suggests is 1 ng RNA input. These results suggest that condensing 10 µL of elute to 5 µL in combination with assuming a 1 ng total RNA input is favourable for successful miRNA library formation.

In contrast, Optimisation 3, which assumed a 5 ng RNA concentration, produced a 2.7× lower yield than Optimisation 2. This lower yield likely explains why previous samples, with an assumed concentration of 10 ng, had poor results; the ratios of reagents and the PCR amplification cycles were incorrect for the sample. Optimisation 4 had the highest miRNA library concentration at 8.19 ng/µL. This, however, required multiple RNA extractions to obtain 20 µL total RNA elute prior to condensing, potentially requiring the pooling of samples to increase plasma volume. Examples of the Fragment Analyzer graphs of the three optimisations (Figure 6a–c) highlight how the protocol adjustments form a purer miRNA library at approximately 200 bp, with considerably less prominent unbound adaptor peaks at approximately 50–60 bp.

### 2.3. Optimisation 2 in Practice

Using the information gathered in Optimisation 2, ninety-two paediatric plasma samples of 100 µL volume underwent miRNA library preparation with the QIAseq miRNA library kit with these protocol adjustments. The samples were split into eight batches, evenly split between the investigation groups, gender, and age to reduce the batch effect. Each batch underwent pre-sequencing qPCR checks with the spiked-in UniSp6 to ensure neutralised levels of RNAse activity during RNA extraction (Figure 7).

The adaptations included within Optimisation 2 appeared to show promising improvements for most of the samples, with an average miRNA library concentration of 5.6 ng/µL (Figure 8 and Figure 9). There is improvement in miRNA library concentration from even the average Optimisation 2 results, with an average concentration of 1.6 ng/µL to an average concentration of 5.6 ng/µL. This further indicates that the paediatric plasma samples extracted at 100 µL are much closer to 1 ng in RNA concentration in the initial starting volume of plasma and that the optimisation steps were indeed beneficial to this sample type. The QIAseq kit handbook assumes a total RNA concentration of 10 ng in serum/plasma, which decreases the reagent dilutions from that of an assumption of 1 ng. However, in this case, those assumptions proved to be less advantageous.

It was determined that a minimum concentration threshold of ≥0.5 ng/µL would be required for viable libraries. As a result, 8 out of 92 samples failed to produce a viable library. The reporting of failure rates in miRNA library preparations is uncommon, making it difficult to assess whether the 8.7% failure rate observed in this case (Figure 10) is within the acceptable range for the field. However, given the challenges associated with generating miRNA libraries from low-concentration, low-volume paediatric plasma samples, this outcome of a 91.3% success rate represents a notable improvement over previous efforts and supports further successful research in this field.

The range of miRNA library concentrations across all the investigated groups (viral, bacterial, inflammatory, and healthy control) is fairly evenly distributed, with similar mean values of approximately 5 ng/µL (Figure 11a). This suggests that the optimisation steps are robust enough to accommodate potential variations in miRNA concentrations across the different groups in these paediatric samples. The spread of miRNA concentrations per batch (Figure 11b) was distributed consistently within an individual batch, but the range varied across different batches, highlighting the batch effect on the samples.

Further to the observation of a concentration range between samples, when looking at individual samples regardless of batch (Figure 12), the group is very heterogeneous. This is most likely due to individual variations in the sample miRNA concentration, but to ensure there were no age or gender differences in the total RNA concentration in the sample that could affect the library formation, this was investigated (Figure 13). No distinct clustering of ages was found within the separated age groups (0–5 years, 5–10 years, and 10–16 years). The majority of samples from all age groups cluster around 0–6 ng/µL, suggesting that this should be an expected result when following these optimisations in this sample type. This also suggests that when investigating paediatric plasma, there should be little to no impact on library formation between different age groups. These results may indicate that although there is sample variation in total input RNA, this is not due to age, and the minimum total RNA concentration required for sequencing is available regardless of age (Figure 13a). To the same extent, gender seemingly has no impact on the ability of this library preparation kit with modifications to produce a highly viable library (Figure 13b).

It was noted during some library preparations, even with optimisation, that there was still a prominent peak in the Fragment Analyzer reports at approximately 50–60 bp, which are unbound adaptors. The most favourable Fragment Analyzer graph should appear as it does in Figure 14a, with only one singular peak at approximately 200 bp. However, in 26% of the samples, a second prominent peak was identified at approximately 50–60 bp, as shown in Figure 14b. Although these peaks are evidence of unbound adaptors, which cannot bind to the flow cell, they do alter the total concentration of the sample, which interferes with the balancing of the library pools prior to sequencing. It was found through trial and error that during the magnetic bead washing steps, rotating the tube three times at each step reduced the presence of the unbound adaptors significantly, so this practice should be routinely incorporated.

## 3. Discussion

There are significant challenges in generating viable small RNA sequencing libraries with paediatric plasma samples, which may impact the progression of knowledge surrounding miRNA in paediatric disease. This project aimed to resolve some of these issues in generating viable libraries with low-volume and low-concentration samples to facilitate further paediatric miRNA research. As there is no consensus on the best RNA extraction kit to use for plasma samples for small RNA sequencing, two kits with different RNA isolation methods were considered. The miRNeasy Serum/Plasma kit utilises QIAzol reagent and column-based methods to extract RNA specifically from low-abundance biofluids such as plasma. Whereas the MagMAX miRVana kit, which is intended to be used via the KingFisher but can be used manually, utilises magnetic bead technology on multiple sample types, including plasma. The miRNeasy Serum/Plasma kit has been reported to outperform other plasma-specific RNA extraction kits for low-volume samples [19,20]. The MagMAX miRVana kit has previously been used for low-volume serum transcriptomics studies [21] and miRNA isomiR analysis [22].

Although the miRNeasy kit is specifically optimised for use with serum/plasma samples, in practice, it showed no significant difference in aiding miRNA library formation compared to the more general MagMAX extraction kit. The performance of both RNA extraction kits in combination with the QIAseq miRNA UDI Library kit, even with condensed total RNA elute as the input material, was poor with an average miRNA library concentration of 0.301 ng/µL and 0.027 ng/µL for the miRNeasy and MagMAX, respectively. The results indicated that optimisation of the QIAseq library preparation was required for these samples. A moderate increase of 1.3 × 10^7^ in average total reads and 4% UMI reads in the miRNeasy extracted samples urged for this kit to be selected for future sample preparations and optimisations. It is, however, possible that the optimisations to the QIAseq protocol would also be beneficial with RNA extracted with the MagMAX kit, and future research will hopefully answer this.

By halving the input material volume, it is expected that you would also halve the input miRNA availability. However, no significant differences were found between using 100 µL or 200 µL starting volumes of plasma in either the total reads or the percent UMI reads. This may indicate that there is enough miRNA present in the sample, even at 100 µL, that can be amplified during the PCR cycles of the library preparation to enable sequencing. There was an outlier result within the percent UMI reads for samples extracted with 200 µL of plasma. This could indicate that reducing the starting volume of plasma reduces the number of possible unique miRNA associated with a sample, thereby reducing the discovery ability. In practice, with the optimised alterations to the QIAseq protocol, the miRNA libraries produced from samples extracted at 100 µL exceed the quality and concentration of samples following the recommended instructions with 200 µL of plasma. An average miRNA library prepped with the Optimisation 2 for 100 µL of plasma had a concentration of 5.6 ng/µL, with an observed maximum yield of 24.3 ng/µL. In contrast, without optimisation, the observed yield using 200 µL of plasma had a maximum concentration of 0.3 ng/µL, and 100 µL had a maximum concentration of 1.4 ng/µL. These data suggest that using a 100 µL starting volume of plasma instead of 200 µL may be beneficial in all cases, especially with the suggested optimisations.

Another challenge with circulating miRNAs is the lack of a consensus on suitable normalisation housekeeping miRNAs [23]. As a result, cohort-specific housekeeping genes must be identified for each study, often requiring techniques such as small RNA sequencing to achieve accurate normalisation. This is particularly true for paediatric circulating miRNAs, where there is limited literature. Housekeeping miRNAs are also useful for sample quality control, including checking for high levels of RNAse degradation, which can inhibit successful library preparation. Therefore, a known quantity of a spike-in exogenous RNA, UniSp6, was used as an alternative. High levels of RNAse in the sample would affect endogenous and exogenous RNA alike, and high Ct values and large variation indicate if a sample could be degraded. This is a particularly important step if planning to pool RNA elutes together; however, even in individual samples, it is best practice to avoid wasting library preparation reagents on a degraded sample. The data suggest that only a limited number of samples had high RNAse levels that were not neutralised during RNA extraction. Initially, one or more samples in the under-90-day-aged pooled group degraded the whole pool, rendering library preparation unusable, and one out of ninety-two individual samples were identified in later analysis.

In theory, the most likely cause of failure to produce a sufficient library is due to the lack of adequate miRNA in the sample. Therefore, any optimisation that increases the concentration of the starting material will benefit the outcome. This can be achieved by concentrating the total RNA elute from a single RNA extraction or combining multiple RNA extraction elutes together and concentrating this down to the input volume of 5 µL. A challenge whilst working with circulating miRNAs is that they cannot be quantified once extracted, as their concentration is too low, and results are unreliable [24]. This challenge means that finding the correct library preparation reagent ratios for the concentration of RNA in the sample is difficult without trial and error for the particular samples used. If there is any leftover adaptor, this can increase the number of adapter dimers that can interfere with sequencing; however, with such a low starting concentration of total RNA, extra PCR amplification steps may be required to boost miRNA library concentration. It may be necessary to prioritise greater yields and potentially require further bead clean-ups to generate a pure sample than to have a poor yield that is pure or still requires further bead clean-ups.

Here, more dilute adaptors and RT primers were required to improve the efficiency of the reactions during library formation. As paediatric plasma is so low in RNA concentration, the library prep reactions were most likely oversaturated with adaptors and RT primers, leading to non-specific ligations and reduced adaptor–miRNA ligations. By reducing the concentration of library preparation reagents, in line with the concentration of RNA and increasing reaction times and cycles, a greater number of beneficial ligation reactions could take place and produce a higher yield. The significance of the effects of altering the reagent ratios was highlighted in Optimisations 2 and 3. Optimisation 2 assumed an input RNA concentration of 1 ng, which changed the ratios of reagents to 1:20/1:10 and increased the PCR amplification cycles to 24. Whereas Optimisation 3 assumed an RNA concentration of 5 ng, which changed the ratios of reagents to 1:10/1:5 and had 22 PCR amplification cycles. Optimisation 3 produced a lower yield of 2.7× less than Optimisation 2, suggesting that the input RNA concentration was closer to 1 ng than 5 ng. This conclusion was validated by the further 92 samples, which were processed via Optimisation 2 reagent ratios and PCR amplification cycles, where the average miRNA library concentration increased by 4 ng/µL from the test Optimisation 2 average.

Investigation groups in this project included aetiologies such as bacterial and viral infection, an inflammatory group, and healthy controls. The generated miRNA libraries from these groups showed even distribution in miRNA library concentrations, all of which had an average concentration of approximately 5 ng/µL, suggesting that with Optimisation 2, the QIAseq kit is robust in different cohorts. In future research, confirming this is true for other diseases/conditions would also be beneficial. While the distribution of investigation groups was consistent, the batch effect was still evident in these samples. The batch effect in miRNA library preparation for sequencing refers to the common systematic, non-biological variations introduced when samples are processed in different experimental runs [25]. Within the eight batches consisting of ninety-two samples, there was a range of average library concentrations of 2.84 ng/µL–11.24 ng/µL. It is essential to spread sample investigation groups, ages, and genders evenly across the batches to reduce the batch effect impacting sequencing and causing misinterpretation of the results.

Although a selection of the miRNA library concentrations was significantly higher than others, the majority of the samples passed the threshold concentration for sequencing. The failure rate observed with the optimised protocol was 8.7%, with 8 out of 92 samples failing to produce a viable library for sequencing. This was determined by the Fragment Analyser miRNA library concentration being ≥0.5 ng/µL. As 26% of the samples prepared with the optimised QIAseq protocol still had a significant peak at 50–60 bp, a second bead clean-up was required to remove the leftover unbound adaptors. Other samples with a less significant peak (50–60 bp peak less than 25% of the height of the 200 bp peak) were adjusted prior to pooling the libraries. This was calculated using the percentage of the miRNA library in the total fragment via the Fragment Analyser smear analysis and multiplying by the Qubit concentration (i.e., Qubit concentration = 2.56 ng/µL, Library % of total fragment = 78.5%, adjusted Qubit concentration = 2.01 ng/µL). The adjusted Qubit concentrations can then be used to equalise all samples in the library pool. Highly concentrated samples can be diluted to the minimum required concentration to ensure balanced pooling before sequencing. Properly balancing the final library pool is essential to guarantee equal representation of samples during sequencing, which is critical for accurate differential expression analysis.

Following the initial failure of all samples in the under 90 days old group, it was considered that young children might have miRNA concentrations in their blood that are too low to support successful miRNA sequencing with the QIAseq kit. Once the QIAseq protocol had been altered to be optimised to work with lower volumes and concentrations of RNA, this theory was quickly discarded. Within the 92 samples used with Optimisation 2, there were 49 children included under 5 years old, 26 children between 5 and 10 years old, and 17 children aged from 10 to 16 years old. No distinct clustering by age was observed within the separated age groups. Most samples across all age groups clustered around 0–6 ng/µL, indicating that this range is typical when following these optimisations for this sample type. This suggests that age has little to no impact on library formation when working with paediatric plasma. There may be distinct miRNA profiles associated with age or gender, with potential variations in expression levels or even the presence of entirely different miRNAs. Nonetheless, age or gender should not lead to a significant reduction in the overall quantity of miRNA. Regardless of age or gender, there should be a sufficient miRNA presence to produce a robust library, although the specific types of miRNAs may vary. If the minimum threshold of total RNA concentration required for library preparation is present, the process will proceed successfully. Variation in miRNA library concentration is also significantly influenced by pre-analytical factors, including sample storage conditions, freeze–thaw cycles, RNA extraction efficiency, haemolysis, and library preparation techniques [13]. These factors often introduce greater variability than the biological differences among individuals [9].

## 4. Materials and Methods

### 4.1. Plasma Samples

#### 4.1.1. FIDO Cohort

300 plasma samples were collected from UK and Irish hospitals for the FIDO Study [26]. All infants aged 90 days and younger presenting with fever (≥38 °C) were included. Plasma sample volumes were limited to 100 µL and have bacterial and viral aetiology. During routine phlebotomy, blood samples were collected and centrifuged at 1500× *g* for 15 min within one hour of sample collection. Frozen samples were then transferred to Queen’s University Belfast. Aliquoted samples were stored at −80 °C until required.

#### 4.1.2. Rapid-19 Cohort

74 plasma and serum samples from children aged up to 16 years were collected from the Royal Belfast Hospital for Sick Children. These samples were of bacterial, viral, or inflammatory aetiology. An additional 200 healthy control plasma and serum samples were collected from Belfast, Cardiff, Glasgow, London, and Manchester from children aged up to 15 years old [27].

Venous blood samples were collected into a BD Vacutainer (Becton, Dickinson and Company, Franklin Lakes, NJ, USA) EDTA tubes by clinical staff. Where possible, blood sampling was timed with routine phlebotomy events to minimise the number of phlebotomy events performed for each child. The venous blood sample was centrifuged at 1500× *g* for 15 min within one hour of sample collection. Aliquoted samples were stored at −80 °C until required.

### 4.2. Testing RNA Extraction Kit and Starting Volume of Plasma in Optimisation 1

#### 4.2.1. RNA Extraction

Paediatric plasma samples from healthy controls, Rapid-19 samples with inflammatory aetiology, and children younger than 90 days old from the FIDO cohort were selected. Due to limited plasma availability, samples were pooled as follows: healthy controls from 3 individuals, inflammatory from 3 individuals, and <90 days from 11 individuals, to produce a pool in each group consisting of 1800 µL. Six pools of paediatric plasma samples underwent miRNA extraction with the miRNeasy Serum/Plasma kit (Qiagen, Hilden, Germany) at a starting plasma volume of either 100 µL or 200 µL. The same plasma pools also underwent manual RNA extraction with the MagMAX mirVana Total RNA Isolation Kit (ThermoFisher Scientific, Waltham, MA, USA) at a starting plasma volume of either 100 µL or 200 µL. The elution volume for the miRNeasy kit is 12 µL, and the MagMAX is 20 µL.

#### 4.2.2. Library Preparation and Sequencing

Due to the anticipated low concentration of miRNA within the plasma samples, Optimisation 1 was assessed. A volume of 10 µL of each sample was condensed with the Savant SPD1010 SpeedVac Concentrator (ThermoFisher Scientific) until dry and resuspended in 5 µL of RNAse-free water to increase the concentration of RNA. Libraries were then prepared using the QIAseq miRNA Library with the 96 Index Kit and IL UDI-A (Qiagen), according to the manufacturer’s protocol for serum and plasma, assuming a 10 ng total RNA concentration. The 3′ adapter was diluted 1:5, the 5′ adapter was diluted 1:2.5, the RT primer was diluted 1:5, and there were 22 PCR amplification cycles.

miRNA libraries underwent quality control with the 5300 Fragment Analyzer System (Agilent, Santa Clara, CA, USA), with miRNA libraries found at approximately 200 bp. Total RNA concentration was measured using the Qubit™ 4 Fluorometer (ThermoFisher Scientific) and was used to generate a balanced final miRNA library pool prior to sequencing. The samples were sequenced on the NextSeq 2000 (Illumina, San Diego, CA, USA) with the P2 cartridge (Illumina), single-end read at 100 base pairs with a read length of 20 M. The data were analysed using Qiagen’s Gene Globe website and the R coding language.

### 4.3. Assessing Optimisations

#### 4.3.1. RNA Extraction and Pre-Sequencing PCR Checks for Inhibitors

Following the manufacturer’s instructions, 100 µL of plasma from 4 individuals were extracted independently with the miRNeasy Serum/Plasma kit. The miRNeasy kit yields 12 µL of total RNA elute, 1.2 µL of which was used to generate cDNA with the miRCURY LNA miRNA SYBR Green PCR kit for Serum/Plasma (Qiagen). A miRNA spike-in, UniSp6 (Qiagen), was added prior to the cDNA production and the PCR assay for UniSp6 was probed for with qPCR via the LightCycler 480 (Roche, Basel, Switzerland). The Ct values for UniSp6 were assessed, and any outliers were deemed to have significant inhibitor interference and were excluded from further investigation.

The total RNA was pooled once the quality control for inhibitors had been passed. All optimisation attempts had their total RNA condensed with the Savant SPD1010 SpeedVac Concentrator (ThermoFisher Scientific) until dry and resuspended in 5 µL of RNAse-free water to increase the concentration of RNA. Optimisations 2 and 3 were condensed from 10 µL to 5 µL, whereas Optimisation 4 underwent condensation from 20 µL to 5 µL.

#### 4.3.2. Library Preparation

The QIAseq miRNA UDI Library kit (Qiagen) was used to prepare the libraries in all optimisations. The kit handbook offers reagent dilutions and cycle numbers depending on the input RNA concentration ranging from 500 ng to 1 ng. Multiple changes were made to the protocol, including altered reagent ratios, increased 4 °C hold times from 5 min to 10 min, increased bead incubation time from 5 min to 15 min, more thorough bead washing steps (Figure 15), and extra bead drying time from 10 min to 20 min. The ratios of reagents were altered in each optimisation as follows:

#### 4.3.3. Optimisation 2

The 3′ adaptor was diluted 1:20, the 5′ adaptor was diluted 1:10, and the RT primer was diluted 1:20. There were 24 PCR amplification cycles. Assumes a 1 ng total RNA input.

#### 4.3.4. Optimisation 3

The 3′ adaptor was diluted 1:10, the 5′ adaptor was diluted 1:5, and the RT primer was diluted 1:10. There were 22 PCR amplification cycles. Assumes a 5 ng total RNA input.

#### 4.3.5. Optimisation 4

The 3′ adaptor was diluted 1:5, the 5′ adaptor was diluted 1:2.5, and the RT primer was diluted 1:5. There were 22 PCR amplification cycles. Assumes a 10 ng total RNA input.

To thoroughly wash the beads, the Eppendorf tube must be rotated 180° 3 times whilst on the magnetic stand before the ethanol removal in each wash step (Figure 15). During the drying stage, the observation of microcracks within the bead pellet indicates the beads are dry; ensure there is also no ethanol remaining on the sides of the Eppendorf tubes (Figure 16). The libraries were created with the QIAseq miRNA UDI Library kit with 96 Index Kit IL UDI-B (Qiagen). The libraries underwent quality control, measuring miRNA library concentration with the 5300 Fragment Analyzer System (Agilent).


Figure 15How to effectively wash the magnetic beads with 80% ethanol. With the beads on the magnetic stand, add 80% ethanol and rotate the tube by 180°. Do this twice more, and then remove the ethanol from the tube. Initially, the beads will move slowly, but with each rotation, they will move more freely through the ethanol. This whole process should then be repeated with fresh 80% ethanol prior to moving on to the drying stage. Created in BioRender by the author Waterfield, T. https://BioRender.com/p05j982 (accessed on 1 February 2025).
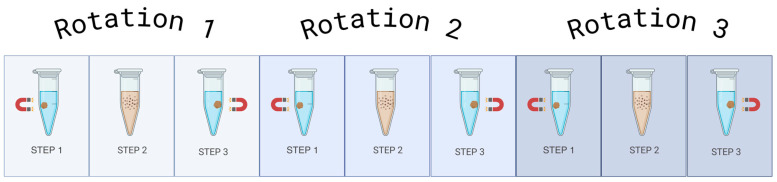

Figure 16Drying step after ethanol bead washes. (**a**) Beads after the ethanol wash step. The beads appear glossy when they still have ethanol present in the Eppendorf tube. This could lead to ethanol carryover, which would impact the yield of the library preparation. (**b**) Dried beads after the ethanol wash step. When the beads are fully dry, they will appear matte and have the presence of microcracks. The beads appearing matte with no microcracks indicate traces of ethanol are still present. All samples must appear matte and with microcracks before the next stage in the protocol can be performed. Extra caution should be taken that no ethanol remains on the sides of the Eppendorf tubes.
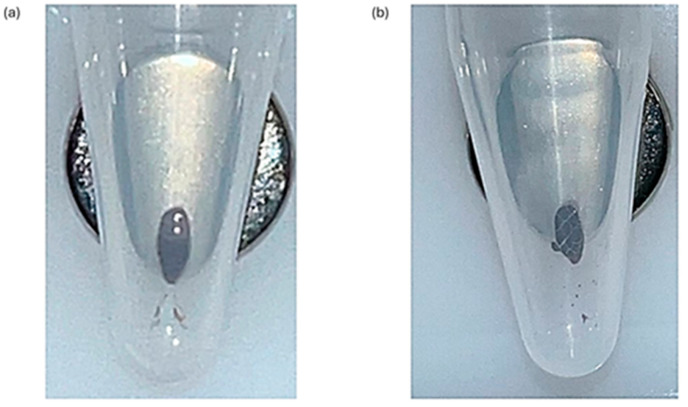



### 4.4. Optimisation 2 in Practice

During this study, 92 paediatric plasma samples from 4 investigation groups (bacterial, viral, inflammatory, and healthy controls) from the FIDO and Rapid-19 cohorts underwent RNA extraction using a 100 µL starting volume of plasma with the miRNeasy Serum/Plasma kit (Qiagen), following the manufacturer’s instructions. The samples were split into 8 batches, evenly split between the cohorts, investigation groups, gender, and age to reduce the batch effect. Each batch underwent pre-sequencing PCR checks via the LightCycler 480 (Roche) with the spiked-in UniSp6 (Qiagen) to ensure neutralised levels of RNAse activity during RNA extraction had occurred. Samples passing quality control had their total RNA condensed until dry with the Savant SPD1010 SpeedVac Concentrator (ThermoFisher Scientific) from 10 µL and resuspended in 5 µL RNAse-free water. The library preparation steps from Optimisation 2 were followed, and the protocol is given in full detail in Appendix A.

## 5. Conclusions

Working with challenging sample types such as plasma adds inherent issues when using techniques such as small RNA sequencing. To add further complexity, working with paediatric plasma at small volumes, such as 100 µL, makes the challenge even greater. While using the QIAseq miRNA UDI library kit, issues with library formation were found at 100 µL and at the recommended 200 µL starting volume of plasma. It was deduced that the issue with the poor miRNA library formation from the samples tested was due to very low endogenous RNA concentration in the plasma. The only possible ways to improve the outcomes were to either increase the concentration of RNA (by adding more volume or condensing the RNA elution) or adapt the protocol to improve the reactions. It was decided that using both methods would be the most effective. Optimisation 2 was the best option for these samples and was used to process a further 92 samples.

With the modifications to the protocol, 88 out of 92 samples (91.3%) were successfully processed, representing a significant achievement given the complexity and low volume of the sample type. Prior to significant optimisation, observed miRNA library concentrations averaged 0.3 ng/µL, and with Optimisation 2, this average grew to 5.6 ng/µL, with an observed maximum yield of 24.3 ng/µL. Initially, it was thought that very young infants may have too low a concentration of miRNA to be sequenced with the QIAseq kit; however, the results suggest that the formation of a viable miRNA library was independent of the child’s age or gender. The miRNA library concentration across all prepared samples was highly heterogeneous, indicating sample variation in total RNA present in each case, but generally this level is above baseline thresholds for producing a successful library with these protocol modifications. These data provide strong evidence that miRNA sequencing from low-volume, low-concentration paediatric plasma samples is viable, paving the way for promising future research opportunities.

## Figures and Tables

**Figure 1 ncrna-11-00011-f001:**
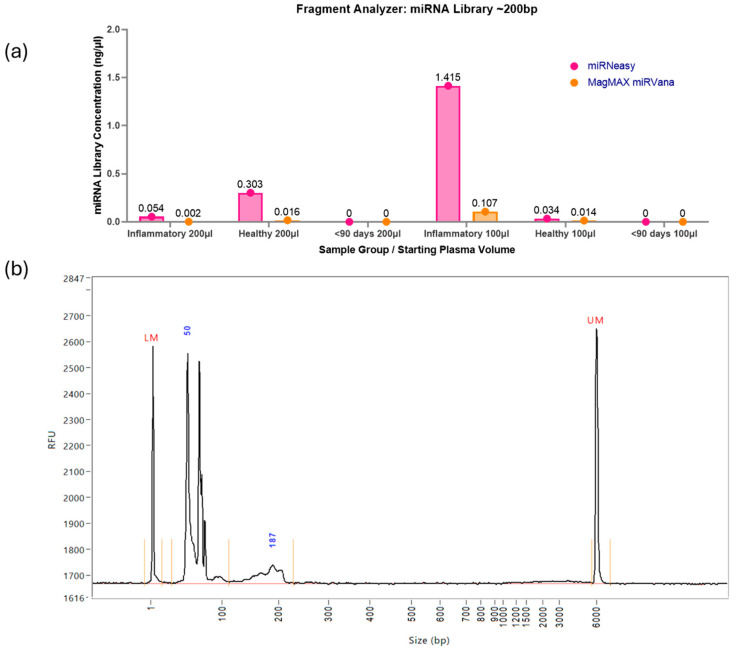
(**a**) Fragment Analyzer miRNA library concentrations. The range of concentrations for these library preps is 0–1.42 ng/µL, indicating a poor yield for the QIAseq miRNA UDI Library kit for paediatric plasma samples when following the manufacturer’s instructions with the addition of condensing the total RNA input. Figure created using GraphPad Prism version 10.4.1 for Windows, GraphPad Software, Boston, MA, USA. (**b**) An example of the Fragment Analyzer Report. There is a prominent peak at approximately 50–60 bp, indicating large quantities of unbound adaptors in the sample, and a minimal miRNA library peak at approximately 200 bp. Each sample includes lower and upper markers to align it with the ladder, and regions are defined with vertical orange lines which are manually applied based on the library profile. The peak size of each region is displayed in blue.

**Figure 2 ncrna-11-00011-f002:**
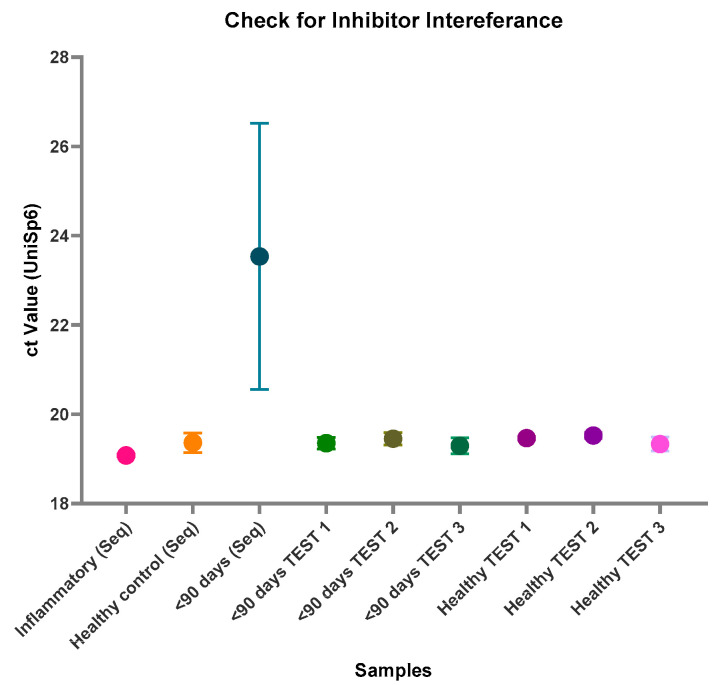
PCR assessment of inhibitors. All pooled samples that were assessed via small RNA sequencing and additional individual test samples were assessed via qPCR and probed for a spike-in quality control, UniSp6. Samples that passed inhibitor quality control have Ct values below 20 and minimal variation. The large variation in the Ct value of the pooled <90 days (Seq) sample used for sequencing indicates the presence of a large concentration of RNA inhibitors. Future samples to be sequenced will be analysed prior to sequencing via qPCR and UniSp6 to ensure there are no inhibitors present before the library preparation is performed. Figure created using GraphPad Prism version 10.4.1 for Windows, GraphPad Software, Boston, MA, USA.

**Figure 3 ncrna-11-00011-f003:**
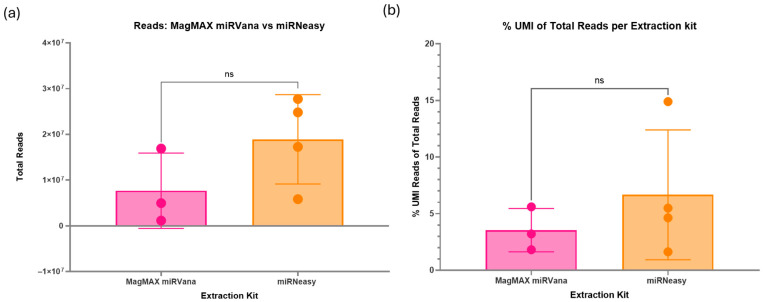
(**a**) Impact of RNA extraction kit on total reads via sequencing. Two RNA extraction kits, the MagMAX miRVana kit and the miRNeasy serum/plasma kit, were assessed for their impact on the small RNA sequencing quality. A paired *t*-test was performed to assess differences between the two methods, with statistical significance set at *p* < 0.05. Data are presented as mean ± standard deviation. There is no significant difference between the results of the two kits. However, one sample from the MagMAX failed to produce a library, so future samples will be prepared using the miRNeasy kit. (**b**) Percentage of unique reads between RNA extraction kits. A paired *t*-test was performed to assess differences between the two methods, with statistical significance set at *p* < 0.05. Data are presented as mean ± standard deviation. The unique reads percentage indicates the number of unique miRNAs within the sample, indicating better discovery ability and quality of the sample. There is no significant difference between the RNA extraction kits; however, the Qiagen miRNeasy kit produced a higher mean percentage. Therefore, moving forward, the Qiagen miRNeasy kit will be used for future samples. Figures created using GraphPad Prism version 10.4.1 for Windows, GraphPad Software, Boston, MA, USA.

**Figure 4 ncrna-11-00011-f004:**
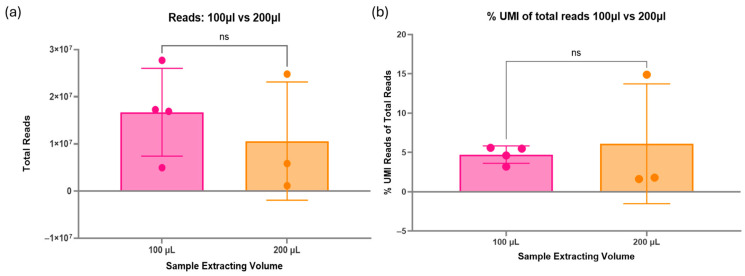
(**a**) Assessment of sample starting volume on sequencing. Due to sample volume constraints in paediatric research, 100 µL was compared to the 200 µL starting volume of plasma. These were the same pooled samples; the only difference was the initial RNA extraction volume. A paired *t*-test was performed to assess differences between the two methods, with statistical significance set at *p* < 0.05. Data are presented as mean ± standard deviation. There was no significant difference between the two volumes on the total reads; however, the 100 µL gave a higher mean of 1.67 × 10^7^. (**b**) Percentage of unique reads between starting volumes. A paired *t*-test was performed to assess differences between the two methods, with statistical significance set at *p* < 0.05. Data are presented as mean ± standard deviation. There was no significant difference between the starting volumes tested, with mean percentages of around 4.65%. Figure created using GraphPad Prism version 10.4.1 for Windows, GraphPad Software, Boston, MA, USA.

**Figure 5 ncrna-11-00011-f005:**
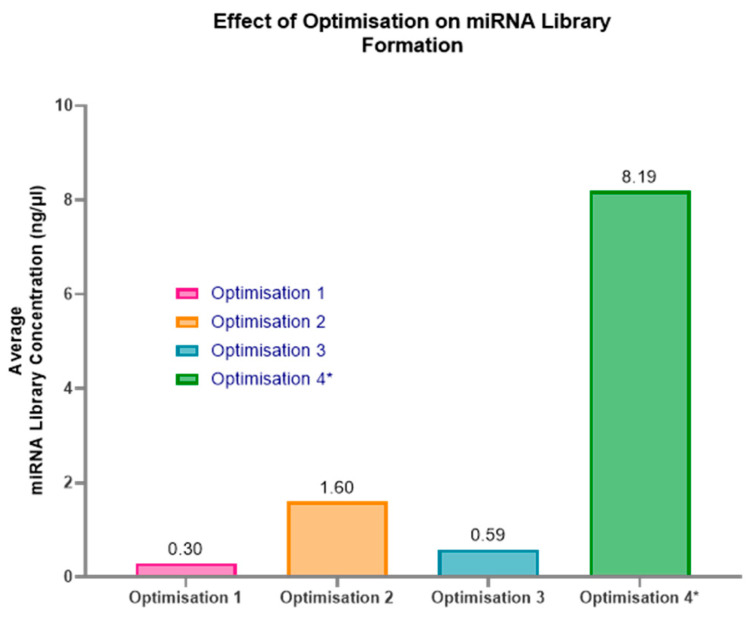
Effect of all optimisation attempts on the formation of miRNA libraries using plasma extracted with the miRNeasy kit. Optimisation 2 is the best methodology for these samples, producing an average miRNA library concentration of 1.6 ng/µL. This methodology assumes 1 ng of total RNA in the sample and requires alterations to the reagent’s ratios (3′ adaptor/RT primer = 1:20, 5′ adaptor = 1:10) and 24 PCR amplification cycles. This suggests that 0.5 ng is in 5 µL of unconcentrated elute from the RNA extraction process using the miRNeasy serum/plasma kit with paediatric plasma samples. * Denotes samples requiring multiple RNA extractions at 100 µL and potential plasma pooling. Figure created using GraphPad Prism version 10.4.1 for Windows, GraphPad Software, Boston, MA, USA.

**Figure 6 ncrna-11-00011-f006:**
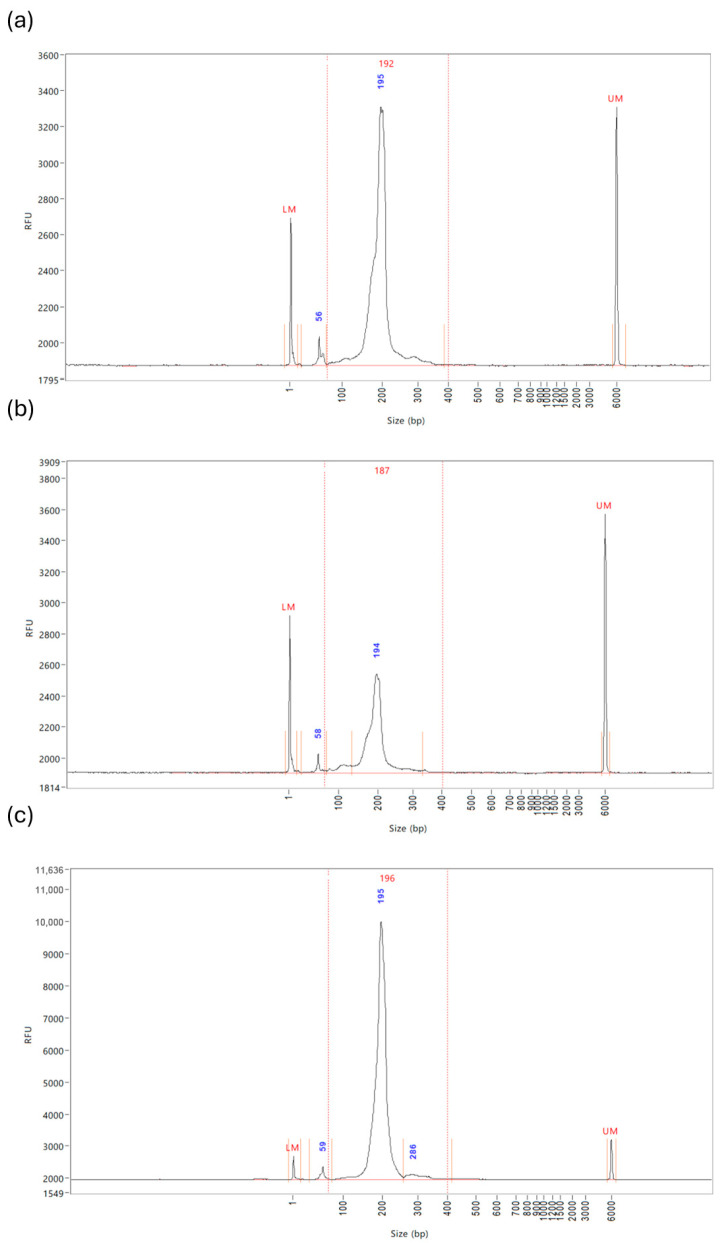
Example Fragment Analyzer graphs of Optimisations 2–4. Each sample includes lower and upper markers to align it with the ladder, and regions are defined with vertical orange lines which are manually applied based on the library profile. The peak size of each region is displayed in blue. (**a**) Optimisation 2—shows minor unbound adaptors with a peak at 50–60 bp at a concentration of 0.055 ng/µL and a significant peak for the miRNA library at a concentration of 1.597 ng/µL. (**b**) Optimisation 3—shows minor unbound adaptors with a peak at 50–60 bp at a concentration of 0.0292 ng/µL, and a moderate peak for miRNA library at a concentration of 0.589 ng/µL. (**c**) Optimisation 4—shows minor unbound adaptors with a peak at 50–60 bp at a concentration of 0.155 ng/µL, and the most significant peak for the miRNA library at a concentration of 8.193 ng/µL.

**Figure 7 ncrna-11-00011-f007:**
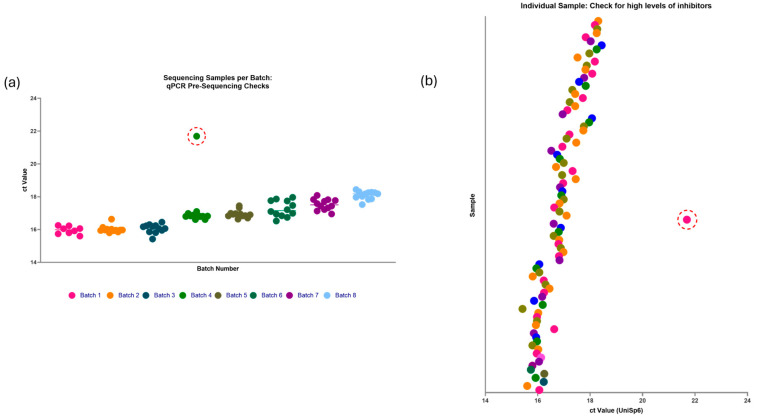
Pre-sequencing checks on ninety-two samples intended to be used with the QIAseq miRNA UDI Library kit with Optimisation 2 protocol. (**a**) Shows individual samples per batch and their Ct value for UniSp6. (**b**) Shows individual samples and their Ct value for UniSp6. The ninety two samples were evenly distributed into batches, with equal numbers from each investigation group in each batch to minimise the batch effect. Out of the eight batches, only one sample (circled in red) had a very high Ct value of approximately 22 for the spiked-in UniSp6, indicating high levels of inhibitors that could interfere with miRNA library formation. This sample was excluded and replaced with a new sample. Figure created using GraphPad Prism version 10.4.1 for Windows, GraphPad Software, Boston, MA, USA.

**Figure 8 ncrna-11-00011-f008:**
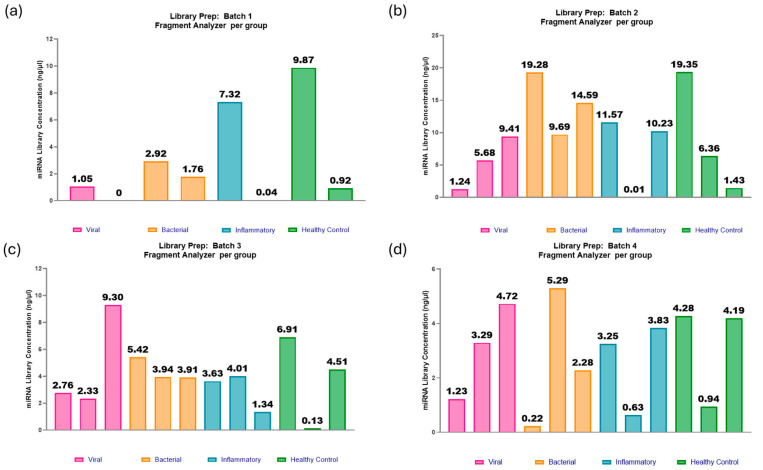
miRNA library preparations: batches 1–4. (**a**) Batch 1: Two out of eight samples failed to produce a viable library, most likely due to ethanol carryover during the wash steps. Of the six samples that produced a library, with a range of concentrations from 0.92 to 9.87 ng/µL and an average concentration of 2.98 ng/µL. (**b**) Batch 2: One out of twelve samples failed to produce a viable library. The range in concentrations was from 0.01 to 19.35 ng/µL, with an average of 9.07 ng/µL. (**c**) Batch 3: One out of twelve samples failed to produce a library. The range in concentrations was from 1.34 to 9.30 ng/µL, and the average was 4.01 ng/µL. (**d**) Batch 4: One out of twelve samples failed to produce a viable library. The range in concentrations was from 0.60 to 5.29 ng/µL, with an average of 2.84 ng/µL. Figure created using GraphPad Prism version 10.4.1 for Windows, GraphPad Software, Boston, MA, USA.

**Figure 9 ncrna-11-00011-f009:**
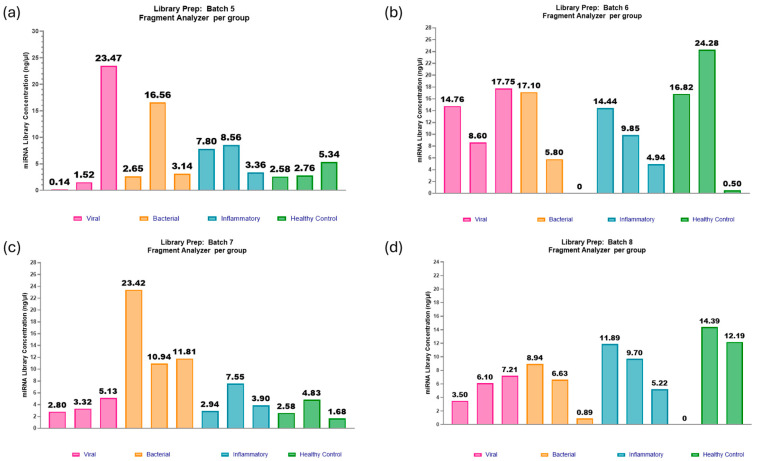
miRNA library preparations: batches 5–8. (**a**) Batch 5: One out of twelve samples failed to produce a viable library. The range in concentrations was 0.14–23.47 ng/µL and an average of 6.49 ng/µL. (**b**) Batch 6: One out of twelve samples failed to produce a viable library. The range in concentrations was 0.50–24.28 ng/µL, with an average of 11.24 ng/µL. (**c**) Batch 7: All twelve samples passed the threshold for miRNA library concentration. The range in concentrations was 1.68–23.42 ng/µL, with an average of 6.74 ng/µL. (**d**) Batch 8: One out of twelve samples failed to produce a viable library. The range in concentrations was 3.5–14.4 ng/µL, with an average concentration of 7.22 ng/µL. Figure created using GraphPad Prism version 10.4.1 for Windows, GraphPad Software, Boston, MA, USA.

**Figure 10 ncrna-11-00011-f010:**
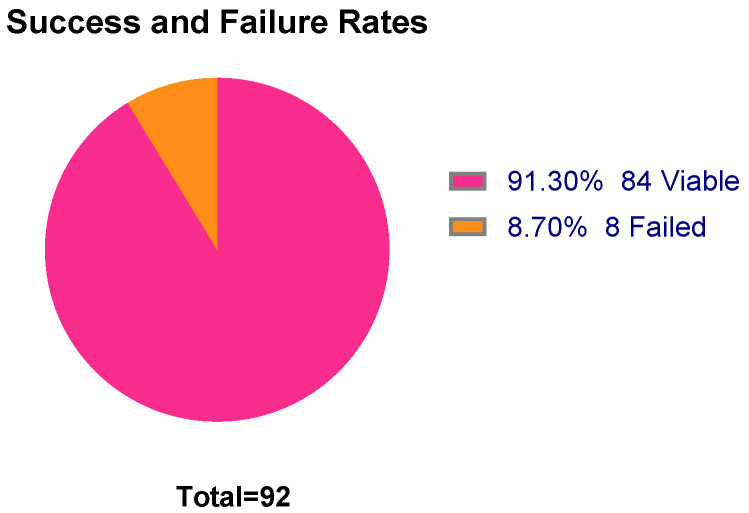
Success and failure rates with the 92 samples and Optimisation 2 methodology. Eight out of ninety-two samples failed to generate a successful library. Eighty-four samples successfully passed quality control, generating a 91.3% success rate. Figure created using GraphPad Prism version 10.4.1 for Windows, GraphPad Software, Boston, MA, USA.

**Figure 11 ncrna-11-00011-f011:**
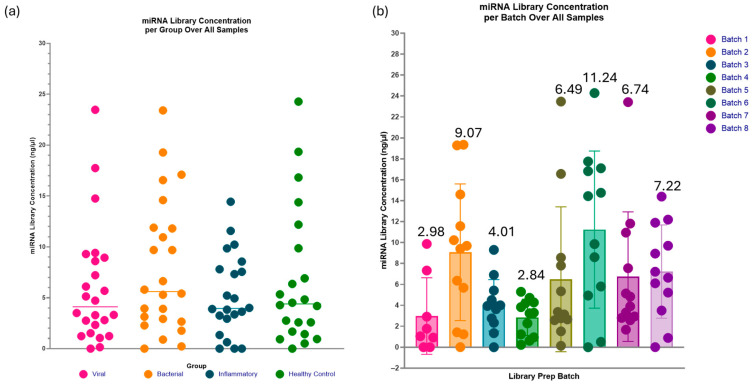
The range of miRNA library concentration per investigated group and batch. (**a**) The spread of miRNA library concentration per investigated group is evenly distributed, with similar mean values at approximately 5 ng/µL. This indicates that irrespective of varying aetiologies and pathophysiological mechanisms, the library generated through the protocol modifications demonstrates comparable performance. (**b**) Evidence of a batch effect is apparent, as the average miRNA library concentrations vary across batches, ranging from 2.84 to 11.24 ng/µL. However, the batch effect should have minimal impacts on sequencing due to the sample groups even distribution between batches and the ability to dilute highly concentrated samples for library pool equalisation prior to sequencing. Figure created using GraphPad Prism version 10.4.1 for Windows, GraphPad Software, Boston, MA, USA.

**Figure 12 ncrna-11-00011-f012:**
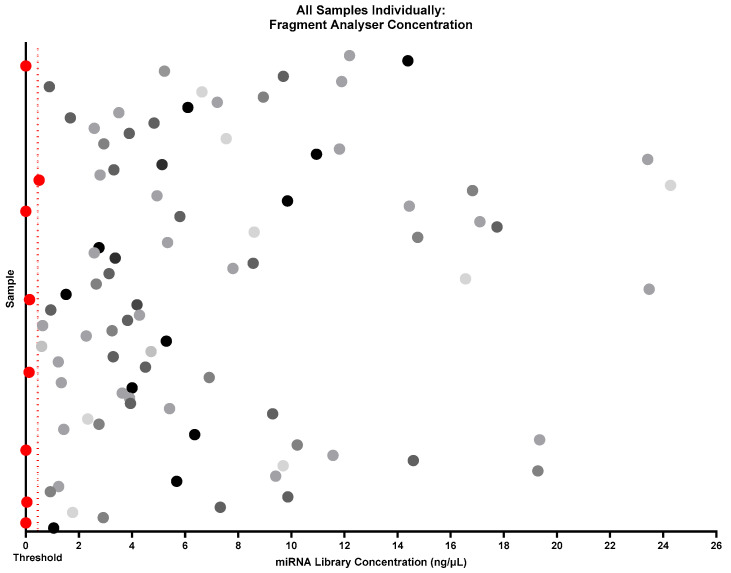
Individual spread of miRNA library concentrations. Samples denoted in red are the samples excluded from sequencing due to being below the threshold concentration of 0.5 ng/µL (marked with a dashed line). The miRNA library concentrations are heterogeneous, with a range between 0 and 24.3 ng/µL and an average concentration of 5.6 ng/µL. Figure created using GraphPad Prism version 10.4.1 for Windows, GraphPad Software, Boston, MA, USA.

**Figure 13 ncrna-11-00011-f013:**
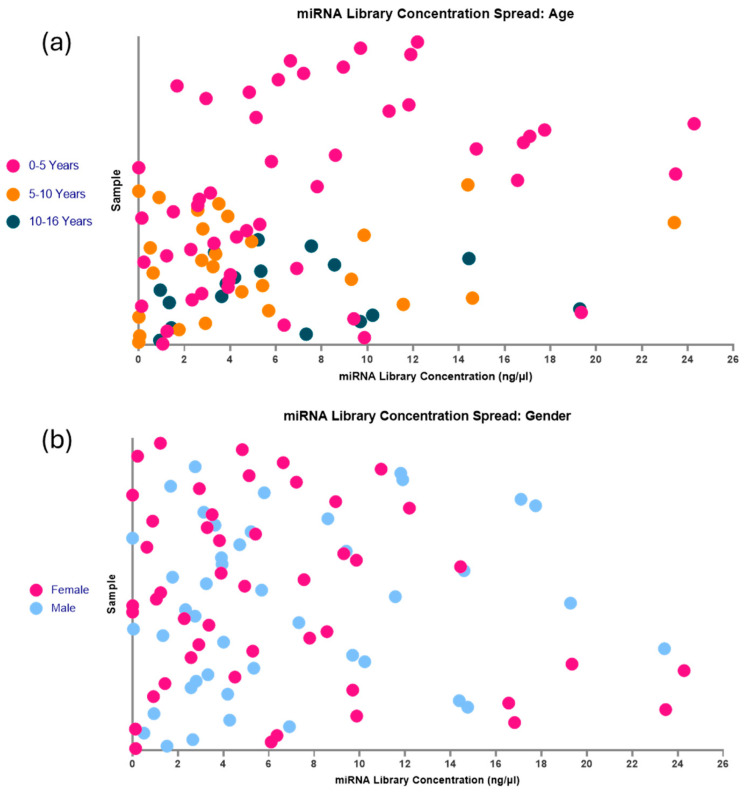
The spread of miRNA library concentrations with age and gender. (**a**) Age: Samples were split into age groups of 0–5 years, 5–10 years, and 10–16 years, and their miRNA library concentrations. No distinct clustering pattern was observed, apart from the average concentration for samples regardless of age at approximately 6 ng/µL. These data suggest that age is not an influential factor in surpassing the minimum total RNA concentration required to prep libraries with the modifications with this kit. (**b**) Similarly, the effects of gender on the formation of a library do not appear to be an influential factor, with a very heterogeneous presentation in these samples. Figure created using GraphPad Prism version 10.4.1 for Windows, GraphPad Software, Boston, MA, USA.

**Figure 14 ncrna-11-00011-f014:**
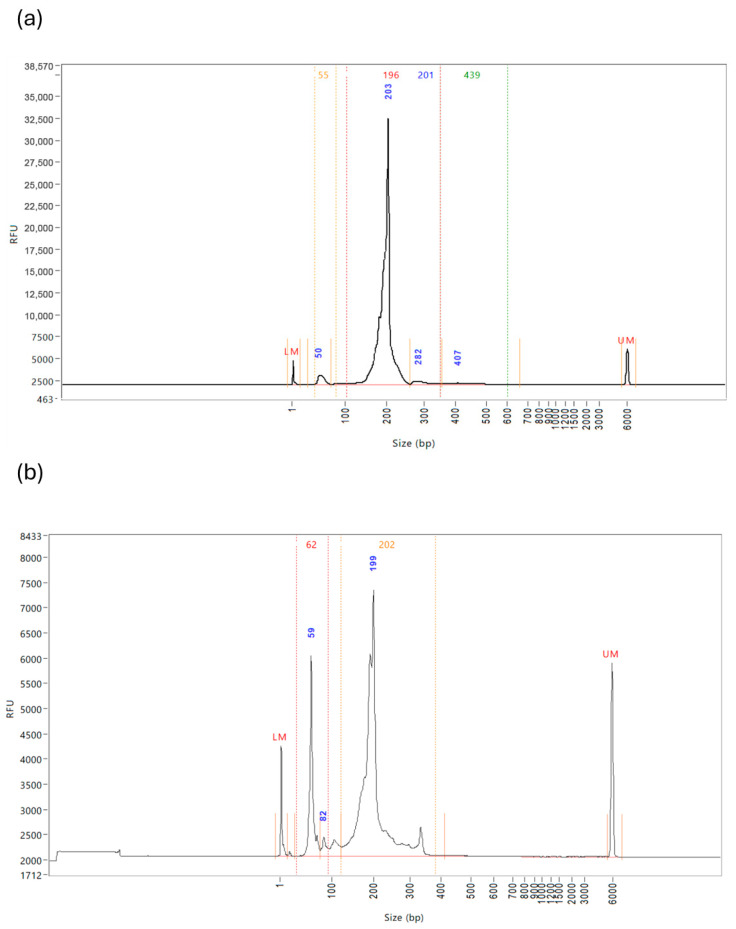
Examples of Fragment Analyzer Reports for miRNA libraries. Each sample includes lower and upper markers to align it with the ladder, and regions are defined with vertical orange lines which are manually applied based on the library profile. The peak size of each region is displayed in blue. (**a**) Pure miRNA Library. There is an observed lack of a peak at approximately 50–60 bp, meaning that minimal unbound adaptors are present in the sample, denoting a pure miRNA library. (**b**) Impure miRNA Library. The observed prominent peak at approximately 50–60 bp shows high levels of unbound adaptor still present in the sample, which requires a further bead clean-up to remove.

## Data Availability

Data are available upon request.

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
