# Peer review of "miRNA Library Preparation Optimisation for Low-Concentration and Low-Volume Paediatric Plasma Samples"

_ncrna, 2025, doi:10.3390/ncrna11010011_

Round 1
Reviewer 1 Report
Comments and Suggestions for Authors
Dear Authors,
Your manuscript addresses a critical challenge in miRNA research: preparing libraries from low-volume pediatric plasma samples. The study's focus on optimizing the QIAseq miRNA UDI Library Kit protocol is both timely and significant.
Your results demonstrate a clear improvement in library preparation, with Optimization 2 offering a practical and effective solution. The high success rate (91%) and the independence from biological factors such as age and gender make your findings particularly compelling. These results not only advance technical workflows but also open new opportunities for miRNA biomarker research in pediatrics.
A few suggestions for improvement:
- While your optimizations are well-explained, adding more context on why specific reagent ratios or PCR cycles were chosen would strengthen the manuscript.
- Summarizing results like success rates, library concentrations, and batch effects in a more visual format would improve accessibility.
- Expanding the discussion of prior work in miRNA library preparation would better contextualize your contributions.
Overall, your manuscript is scientifically sound, novel, and impactful.
Author Response
Dear Reviewer,
Thank you for taking the time to review our paper, it is much appreciated. Below are some responses regarding your comments. I found comments 2 + 3 slightly vauge, so please do let me know if you have something specific in mind and I will do my best to ammend if what we have provided to be unsatisfactory.
Comment 1: While your optimizations are well-explained, adding more context on why specific reagent ratios or PCR cycles were chosen would strengthen the manuscript.
Response 1: Thank you for pointing this out. I have added in that the suggested changes to reagent dilutions and PCR cycles which were given in a table from the protocol handbook.
- Page 3, line 108-109 = "Other reagent dilutions are offered throughout the protocol for 500ng,100ng, 10ng, 1ng of total RNA input."
- Page 5, line 202-203 = "The kit handbook offers reagent dilutions and cycle number depending on the input RNA concentration ranging from 500ng to 1ng."
Comment 2: Summarizing results like success rates, library concentrations, and batch effects in a more visual format would improve accessibility.
Response 2: I am not entirely sure what other figures the reviewer would like to see included. There are figures for library concentrations and batch effect, I have added in a new figure for the success/failure rate. If extra are required please be specific and I will aim to address.
- Page 13, line 437-439 = I have added in a figure summarising success and failure rates of the 92 samples prepared with the optimisation 2 protocol.
- Figure 3a, Figure 7, Figure 10 + Figure 11 = all bar charts for visual representation of library concentrations.
- Figure 9 and Figure 13 = both of these figures visually show the batch effect. Figure 9 shows the PCR batch effect on ct value of a known quantity of spiked in RNA (slight increase in ct value per batch). Figure 13a shows that the batch effect had no influence on the investigation groups, but there was a range of miRNA library concentrations that was influenced by batch (Figure 13b).
Comment 3: Expanding the discussion of prior work in miRNA library preparation would better contextualize your contributions.
Response 3: There have been no other optmisations (as far as I am aware) to the QIAseq kit for plasma samples. The QIAseq kit has been included in at least 2 other assesments comapring optimimum library preparation, and has been shown to produce consistently good results. If the reviewer would like me to discuss something specific please do let me know and I will update further.
- Page 2, 86-87 = "Quantification bias, which is particularly prominent in biofluids due to low miRNA concentration, can occur due to unequal ligation efficiencies leading to over and under estimation of the true miRNA signatures"
- Page 3, line 102-103 = "A systematic assessment of six low input miRNA library preparation kits found the QIAseq kit to reliably demonstrate top performance among those tested".
- page 3, line 104-105 = "Furthermore, a separate assessment of seven commercially available small RNA sequencing library preparation kits for biofluids found the QIAseq kit to perform among the best in all metrics tested"
Reviewer 2 Report
Comments and Suggestions for Authors
The manuscript by Rodgers, Watson and Waterfield sought to enhance the outcome of miRNA library preparation from plasma samples of pediatric patients using the QIAseq miRNA UDI Library kit
They argue, and justifiably so, that sample volume from pediatric patients is low and therefore established the need to optimize existing protocols so as to enhance the generation of viable miRNA libraries.
A number of different strategies were utilized from the outset starting from the choice of RNA isolation method (magnetic bead vs Qiazol/column extraction) to alterations to the miRNA UDI Library kit protocol. Of the protocol alterations that were experimented upon, the second optimization protocol (Optimisation 2) provided the highest yield for individual samples extracted with 100μl of plasma. This protocol made appropriate reagent dilutions with the assumption of a staring concentration of 1ng of RNA.
Taken together, the authors have provided a series of well-tailored, logical protocol alterations which culminated in a marked improvement in the generation of miRNA libraries from pediatric patients.
I have the following minor observations
In the description of RNA extraction, the authors mentioned that samples from healthy controls, an inflammatory cohort of children less than 90 days old were selected. The preceding section on plasma samples did not allude to a particular inflammatory cohort of patients. It may be assumed that the authors were referring to the Rapid-19 cohort, however even this cohort had samples of bacterial, viral and inflammatory sources. The authors should please make this part clearer to understand
The quality of the Fragment Analyzer graphs needs to be improved. In most cases, the values on both the x and y axes are barely legible.
The figure headings do not all have the same font size, and some are more legible than others. A classic example of this is Figure 11, where 11a differs from 11b, c and d
Author Response
Dear Reviewer,
Thank you for taking the time to assess our paper, it is very appreciated. Below are our responses to your comments.
Comment 1: In the description of RNA extraction, the authors mentioned that samples from healthy controls, an inflammatory cohort of children less than 90 days old were selected. The preceding section on plasma samples did not allude to a particular inflammatory cohort of patients. It may be assumed that the authors were referring to the Rapid-19 cohort, however even this cohort had samples of bacterial, viral and inflammatory sources. The authors should please make this part clearer to understand
Response 1: Thank you for picking this up, we have ammended the methods for Optimisation 1.
- Page 4, line 155-156 = "Paediatric plasma samples from healthy controls, RAPID-19 samples with inflammatory aetiology and children younger than 90 days old from the FIDO cohort were selected. "
Comment 2: The quality of the Fragment Analyzer graphs needs to be improved. In most cases, the values on both the x and y axes are barely legible
Response 2: Thank you for picking this up, this has been ammended.
- Figure 3b, Figure 8 and Figure 16 have been made larger and less pixelated.
Comment 3: The figure headings do not all have the same font size, and some are more legible than others. A classic example of this is Figure 11, where 11a differs from 11b, c and d
Response 3: The sub-figures have all been made the same size within one figure (ie Figure 11 a,b,c+d are all equal).